# Insight into the Mammalian Aquaporin Interactome

**DOI:** 10.3390/ijms23179615

**Published:** 2022-08-25

**Authors:** Susanna Törnroth-Horsefield, Clara Chivasso, Helin Strandberg, Claudia D’Agostino, Carla V. T. O’Neale, Kevin L. Schey, Christine Delporte

**Affiliations:** 1Division of Biochemistry and Structural Biology, Lund University, 22 100 Lund, Sweden; 2Laboratory of Pathophysiological and Nutritional Biochemistry, Université Libre de Bruxelles, 1070 Brussels, Belgium; 3Department of Biochemistry, Vanderbilt University School of Medicine, Nashville, TN 37240, USA

**Keywords:** aquaporin, interactome, protein–protein interaction, trafficking, function, mammalian

## Abstract

Aquaporins (AQPs) are a family of transmembrane water channels expressed in all living organisms. AQPs facilitate osmotically driven water flux across biological membranes and, in some cases, the movement of small molecules (such as glycerol, urea, CO_2_, NH_3_, H_2_O_2_). Protein–protein interactions play essential roles in protein regulation and function. This review provides a comprehensive overview of the current knowledge of the AQP interactomes and addresses the molecular basis and functional significance of these protein–protein interactions in health and diseases. Targeting AQP interactomes may offer new therapeutic avenues as targeting individual AQPs remains challenging despite intense efforts.

## 1. Introduction

Aquaporins (AQPs) are a family of transmembrane water channels expressed in all living organisms that facilitate osmotically driven water flux across biological membranes and, in some cases, the movement of small solutes (such as glycerol, urea, CO_2_, NH_3_, and H_2_O_2_) [1,2]. In mammals, 13 different AQPs have been identified that can be subdivided into three groups: classical AQPs permeable to water (AQP0, AQP1, AQP2, AQP4, AQP5, AQP6, and AQP8); aquaglyceroporins permeable to water as well as to glycerol and small solutes (AQP3, AQP7, AQP9, and AQP10); and unorthodox AQPs (AQP11 and AQP12) with lower sequence homology [1,3]. While the solute selectivity of the latter two remains to be conclusively established, AQP11 has been suggested to facilitate the transport of glycerol [4] and H_2_O_2_ across the endoplasmic reticulum (ER) membrane [5].

Structural studies of AQPs from a wide range of organisms have revealed a common fold consisting of six transmembrane helices (TM1-6) and five connecting loops (loops A–E). Loops B and E fold back into the membrane from opposite sides and form shorter helices that constitute a seventh pseudo-transmembrane segment (Figure 1A). A pore runs through the middle of each monomer where water (or other solute) molecules line up in single file. In addition, the assembly of four monomers into a tetramer creates a fifth central pore that has been proposed to conduct gas or other hydrophobic molecules. The tetrameric assembly has been proposed to be primarily mediated by interactions between loop D of the different monomers (Figure 1B,C) [6].

Within the water-conducting pore, two highly conserved asparagine–proline–alanine (NPA) motifs are located at the ends of the short helices formed by loops B and E (Figure 1A,D). Together with the aromatic–arginine region (ar/R), which is located towards the extracellular side of the pore, the NPA region is believed to play a crucial role in preventing proton translocation along the water chain by a Grotthuss-type mechanism. The ar/R region constitutes the narrowest part of the pore and functions as a selectivity filter. Except for a highly conserved arginine, the amino acid composition of the ar/R region differs between the AQP subgroups, thereby providing different solute specificity.

Protein–protein interactions are fundamental for the organization and regulation of mammalian AQPs. The tetrameric assembly of AQPs has been shown to be important for regulation by trafficking and for water permeability. In addition to forming homotetramers, or heterotetramers of splicing variants/mutants, as shown for some AQPs [2], mammalian AQPs form protein complexes with a wide range of interacting partners [7,8]. These interactions are involved in the functional regulation of both AQPs and their interacting partners and play important roles in physiological and pathological processes. Regulatory interactions typically involve the cytoplasmic AQP N- and C-termini, which are less conserved than the protein core and often contain multiple post-translational modification sites that allow AQP-specific regulation in a tissue-dependent manner. In water-specific mammalian AQPs, the proximal part of the C-terminus forms a short amphipathic alpha helix that is known to participate in several interactions with regulatory proteins and thus has been proposed to be a common protein–protein interaction site (Figure 1A,C).

In this review, we provide an in-depth overview of the current understanding of AQP interactomes and address the molecular basis and functional significance of such protein–protein interactions. Considering the large number of AQP-interacting proteins discovered so far, it is difficult to exhaustively discuss all of them. Therefore, our focus lies on the interactions that are best characterized. To the best of our knowledge, no interaction partners have been identified for AQP10–AQP12; therefore, these are not included in the discussion below.

## 2. AQP0

AQP0 is the most abundant membrane protein in the mammalian lens, making up ~50% of the lens fiber cell membrane proteome [9,10,11], and has also been reported to be expressed in the retina [12], testes [13], and liver [14]. In the lens, AQP0 is essential for maintaining lens transparency by functioning as a water channel as well as a cell-adhesion molecule [15]. The latter involves the interaction between AQP0 tetramers in adjacent fiber cells, forming thin junctions that help uphold the compact lens core structure. Junction formation is believed to be induced by the age-dependent proteolytical cleavage of AQP0 N- and C-termini [16] and structural studies have shown that the junctional form of AQP0 is closed [17,18], suggesting that it does not facilitate water transport between cells. Functional studies, however, contest this notion and demonstrated water permeability in truncated AQP0 [19,20].

Several proteins have been reported to interact with the AQP0 C-terminal tail, one of which is calmodulin (CaM). CaM is an extensively studied Ca^2+^-binding protein that interacts with and regulates a wide range of proteins in eukaryotic cells, including scaffolding proteins, signal transduction proteins, phosphatases such as calcineurin, and Na^+^ and K^+^ channels [21]. AQP0 directly interacts with CaM via its C-terminal helix and several studies have characterized this interaction structurally and functionally. Early work by Girsch and Peracchia [22] using liposome swelling assays implicated CaM binding as a regulatory mechanism for decreasing AQP0 water permeability. A low-resolution structure of the AQP0-CaM complex determined using electron microscopy indicates the structural arrangement [23]. This structure was in agreement with previous reports that two CaM molecules bind one AQP0 tetramer, with each CaM interacting with two AQP0 C-termini in an anti-parallel manner [23,24], a process that has been shown to exhibit positive cooperativity [25]. Molecular dynamics simulations have implicated the AQP0 arginine-rich intracellular loop D as a likely allosteric site for its regulation by CaM and it was proposed that this allosteric interaction alters a constriction site at the cytoplasmic end of the AQP0 pore, thereby reducing water permeability [26]. Furthermore, the phosphorylation of AQP0 at Ser 235, a known mechanism for regulating AQP0 water permeability, has been experimentally shown to inhibit CaM binding, adding another level of control [24,27].

AQP0 has been shown to directly interact with A-kinase anchoring protein 2 (AKAP2) [28], a member of the AKAP family that associates with protein kinase A (PKA) to localize the kinase to specific regions within the cell [29]. By serving as a scaffold for PKA and its substrates, AKAP2 indirectly modulates an assortment of cellular processes [29]. AKAP2 was reported to interact with both PKA and the distal C-terminal region of AQP0 in outer cortical lens fiber cells [28]. Immunohistochemistry experiments confirmed the co-localization of AQP0 and AKAP2 in mouse lens sections and immunoprecipitation and SPOT array experiments demonstrated that the C-terminal tail of AQP0 is a specific AKAP2 interaction site [28]. The authors suggested that the sequestration of AQP0 and PKA by AKAP2 facilitates AQP0 phosphorylation at Ser235 and ultimately disrupts its interaction with CaM, thereby modulating AQP0 water permeability [22,28].

Ezrin belongs to the ERM (ezrin, radixin and moesin) protein family that contains an FERM (Four-point-one ERM) domain known to associate with proteins to provide a link to the actin cytoskeleton [30]. This group of proteins play a variety of essential roles in cells, including maintaining cellular structure and adhesion, and cell signaling [30]. Through crosslinking mass spectrometry, co-immunoprecipitation, and AQP0 C-terminal peptide pull-down experiments, it was shown that the C-terminal region of AQP0 interacts directly with the N-terminal FERM domain of ezrin via both F1 and F3 subdomains [31]. Compared to CaM, which interacts with the C-terminal amphipathic helix region closer to the membrane, ezrin associates with the distal C-terminal region of AQP0 [31]. Therefore, it is speculated [31] that AQP0 is a potential transmembrane protein linker to the novel cortex adherens junction complex containing ezrin, periplakin, periaxin, and desmoyokin (EPPD) in the lens, as described by Straub et al. [32].

Intermediate filaments are one of the three main filament components that make up the cellular cytoskeleton [33]. Two intermediate filaments specific to the lens, beaded filament structural protein 1 (BFSP1; also called filensin) and beaded filament structural protein 2 (BFSP2; also called 49 kDa Cytoskeletal Protein (CP49) or phakinin), function to maintain cellular structure and are vital for lens transparency, with mutations of either protein leading to cataracts [33]. In addition to maintaining cellular organization, it has been proposed that lens-specific intermediate beaded filaments in conjunction with protein chaperones manage the extremely high protein concentrations found in the lens cytoplasm to maintain lens transparency [33]. Filensin and CP49 co-assemble [34], but studies have also shown that both proteins can associate with AQP0, an interaction that likely plays a role in membrane/cytoskeleton organization. AQP0 affinity purification and LC-MS/MS experiments showed that filensin interacts with AQP0 [35], while subsequent crosslinking mass spectrometry experiments localized the sites of interaction to the AQP0 C-terminal tail and the truncated C-terminal filensin fragment [36]. Immunoconfocal and immunoelectron microscopy experiments verified that AQP0 and filensin spatially interact in bovine lens tissue sections [35]. Additionally, AQP0 permeability assays using Xenopus oocytes showed that AQP0 binding to the filensin tail region reduces AQP0 permeability [37]. In contrast to the effect of AQP0 phosphorylation on CaM binding (discussed above), it was shown that the pseudophosphorylation of the AQP0 C-terminus peptide did not impact filensin binding [37]. AQP0 affinity purification experiments coupled with LC-MS/MS analysis have also identified CP49 as an interacting partner of the C-terminal tail of AQP0 [35]. However, since CP49 associates with filensin to form beaded filaments, it is unclear if CP49 directly interacts with AQP0 [35].

Gap junctions are transmembrane channels that link neighboring cells together, enabling cytoplasmic molecules, usually under 1 kDa, to freely travel from one cell to the other [38]. Gap junctions consist of connexins (Cxs), a family of membrane proteins, that can join to form macromolecular complexes such as connexons or hemichannels that are composed of six connexin proteins [39]. Early co-immunolabeling experiments of AQP0 and connexin 45.6 (Cx45.6) showed that both proteins colocalized in embryonic chick lens cells, and immunoprecipitation in conjunction with immunoblotting experiments validated this interaction [40]. Furthermore, a pull-down experiment using an AQP0 C-terminus fusion protein suggested that Cx45.6 interacted with the C-terminal region of AQP0 in embryonic lenses [40,41]. More specifically, it was reported that the intracellular loop domain of Cx45.6 directly interacts with the C-terminal region of AQP0 using surface plasmon resonance experiments [41]. The AQP0–Cx50 interaction was also reported to increase Cx50 gap junction coupling due to AQP0′s adhesion properties; however, this interaction was not shown to impact Cx50’s hemichannel function [38]. An AQP0/Cx50 double-knockout mouse model displayed substantial morphological changes, such as alterations of lens size, fiber cell structure and intercellular space, highlighting the critical role these two proteins play in establishing lens fiber cell structure and organization [42].

Lens crystallins are water-soluble proteins that, in mammals, are comprised of three main varieties: α-, ß-, and γ-crystallins. Crystallins are the most abundant protein in the human lens [43]. Confocal fluorescence resonance energy transfer (FRET) experiments confirmed previous reports that AQP0 interacts with lens crystallins (αA-, αB-, ßB2-, and γC-crystallins) with varying affinity [44]. The α-crystallin family functions as small heat shock proteins that bind to partially unfolded proteins to prevent aggregation, an important property for maintaining long-term lens transparency [43]. Studies demonstrated that the thermally induced aggregation of AQP0 was prevented by α-crystallin [45]. Co-immunoprecipitation experiments and yeast two-hybrid screening revealed that γE-crystallin interacts with AQP0 and it was proposed that this interaction involves the AQP0 C-terminal region [46]. Employing confocal fluorescence microscopy, this group also showed that AQP0 recruits γE-crystallin to the plasma membrane [46]. Lastly, an endogenous crosslink was identified between AQP0 and γS-crystallin, suggesting that these proteins directly interact in vivo [47].

Figure 2 recapitulates the identified protein–protein interactions involving AQP0.

## 3. AQP1

AQP1 is widely expressed within the human body [1,3] and has been shown to interact with few proteins.

The interactions of hAQP1 with protein LIN-7 homolog (Lin7), ß-catenin and focal adhesion kinase (FAK) have been shown to play key roles in cytoskeleton rearrangement, cell adhesion and cell migration [48,49]. The latter involves increased ion transport at the leading cell edge, which causes water entry through AQP1, cell swelling and cytoskeleton reorganization, involving actin polymerization [50,51] (Figure 3A). The silencing of AQP1 led to dramatic changes in cytoskeleton organization, resulting in a rounded cell shape, as well as the proteasome-mediated degradation of Lin-7 and ß-catenin [48]. Moreover, such protein–protein interactions may be responsible for the activation of small GTP Binding Protein RhoA (RhoA) and Rac Family Small GTPase (Rac), which are also involved in similar cell functions [52]. The precise domains of each protein interacting with AQP1 remain to be elucidated. In light of their physiological relevance, the AQP1–protein complexes are anticipated to be key players in various diseases, especially in cancer, and to be considered as potential targets for new therapeutic options [53]. Moreover, taking into account that AQPs have been implicated in various cancers [54], it is tempting to speculate that FAK, ß-catenin and LIN-7 may interact with other AQPs besides AQP1, including AQP5, as suggested by indirect experimental evidence [55].

Amyloid precursor protein (APP) has been shown to interact with the N-terminal part of hAQP1 in Alzheimer’s disease brains via its C-terminal fragment, amyloid-beta (Aß) [56]. Moreover, APP increased AQP1 expression in brains from patients with Alzheimer’s disease through an epigenetic mechanism [57] and AQP1 overexpression reduced Aß production arising from APP cleavage by inhibiting the binding between ß-secretase (BACE1) and APP [58] (Figure 3B). It was proposed that the AQP1–APP complex may alter brain water homeostasis by interfering with astrocyte water flow and cell motility [56]. Further studies are required to evaluate this hypothesis.

## 4. AQP2

AQP2 is mainly expressed in kidney collecting duct principal cells where its arginine vasopressin (AVP)-dependent trafficking to the apical membrane is crucial for regulating water reabsorption and urine concentration [59,60]. AQP2 trafficking is tightly controlled by multiple post-translational modifications within the AQP2 C-terminus. Indeed, following AVP release from the pituitary gland, a signaling cascade results in the phosphorylation of the AQP2 residing in storage vesicles at Ser 256 by the cAMP-dependent protein kinase A (PKA). This triggers the fusion of the vesicles with the apical membrane, thereby increasing membrane water permeability [61]. The role of the phosphorylation of sites other than Ser256 is less well established; however, sequential phosphorylation at Thr269 (Ser269 in rodents) and Ser261 dephosphorylation have also been shown to increase AQP2 apical membrane abundance [62]. The ubiquitination of Lys 270 triggers AQP2 endocytosis and sorting into intraendosomal vesicles, from which AQP2 is targeted for lysosomal degradation, where it is degraded in the lysosome or released into the urine as exosomes [63]. Furthermore, Ser264 phosphorylation by PKA, Ser265 phosphorylation by Golgi casein kinase, Ser269 phosphorylation by non-receptor tyrosine kinase Src, cyclic GMP-dependent protein kinase G (PKG) and intracellular calcium mobilization are also involved in AQP2 trafficking to the plasma membrane, while protein kinase C is involved in AQP2 endocytosis [59].

Considering the physiological importance of AQP2 in renal physiology and pathophysiology, the AQP2 interactome has been extensively explored. Indeed, a wide range of rat AQP2-interacting proteins (139 proteins) have been identified by cross-linking, immunoprecipitation and LC-MS/MS analysis of the immunoprecipitated material [64]. When comparing the AQP2 interactome with the urea transporter A1 (UT-A1) interactome, some AQP2-interacting proteins (53 proteins) were considered to be proteins playing housekeeping functions common to all membrane proteins processed through the rough-endoplasmic reticulum/Golgi/endosomal pathway, while others (86 proteins) were considered as specifically interacting with AQP2 [64]. These AQP2-interacting partners were associated with various biological processes and molecular functions, such as cell adhesion and small GTPase function [64]. Other studies have identified additional AQP2-interacting partners. Therefore, considering the high number of identified AQP2-interacting partners, only a few of them involved in regulating AQP2 endocytosis/exocytosis will be discussed below in more detail and are summarized in Figure 4.

AQP2 endocytosis/exocytosis involves actin skeleton reorganization and other downstream effectors such as Rho GDP-dissociation inhibitor (RhoGDI) and Ras Homolog Gene Family, Member A (RhoA) [65]. A yeast two-hybrid screening of a human kidney cDNA and co-immunoprecipitation analysis have shown Heat shock cognate 71 kDa protein (Hsc70) and Heat shock protein of 70 kDa (Hsp70) (which share considerable sequence homology, structures and functions) to interact with the AQP2 C-terminus [66,67,68]. The AQP2 Ser256 residue was proposed to be important for this interaction and binding was reduced upon AVP-stimulated phosphorylation at this site [66,68]. It has been proposed that Hsp70 acts a chaperone for AQP2 and tethers murine double minute 2 (MDM2) E3 ubiquitin ligase, which is involved in AQP2 degradation [68]. Studies conducted on mice have revealed that signal-induced proliferation-associated 1 like 1 (Sipa1l1) binds the cytoplasmic PDZ motif of AQP2 and accelerates its endocytosis in the absence of AVP. The phosphorylation of AQP2 Ser269 following AVP stimulation disrupted the binding of Sipa1l1 and consequently prolonged AQP2 retention in the plasma membrane [69]. In addition, the signal-induced proliferation-associated gene-1 (Spa1), a GTPase-activating protein involved in sorting membrane proteins, was also found to interact with AQP2 and to play a role in rat AQP2 trafficking to the apical plasma membrane [70]. SPA-1 binding to AQP2 may reduce the levels of Ras-related protein 1 (Rap1)-GTP, which triggers F-actin disassembly, resulting in the promotion of the AQP2 sorting [70]. Other proteins, such as the synapse-associated protein-97 (SAP97) binding to the C-terminus of human AQP2 [71] and A-Kinase anchoring protein 220 (AKAP220) binding to mouse AQP2, may also regulate AQP2 phosphorylation [72]. It has been suggested that activated PKA, after AVP stimulation, is bound to AKAP and that the AKAP/PKA complex is tethered to a PDZ domain located in the AQP2 C-terminus through SAP97 [71,73]. In addition, SAP97 and Spa1 may play complementary roles, whereby SAP97 may play a role in PKA localization to phosphorylate AQP2 on Ser256 residue, whereas the interaction of Spa-1 with the phosphorylated AQP2 Ser256 residue may be involved in cytoskeleton dynamics during AQP2 translocation to the plasma membrane [71].

Both θ and ζ members of the highly conserved 14-3-3 protein family, which are known to interact with and modulate the function of target proteins in eukaryotes, have been shown to interact with the mAQP2 C-terminus [74]. The binding of 14-3-3ζ to AQP2 relies on the phosphorylation of Ser256 and/or Ser269, and was proposed to prevent or limit the binding of E3 ligase or an adaptor protein, resulting in reduced AQP2 ubiquitylation, internalization and subsequent degradation [74]. In contrast, the binding of 14-3-3θ to AQP2 was suggested to recruit a protein phosphatase, resulting in AQP2 dephosphorylation and internalization [74]. The ubiquitin-specific peptidase 4 (USP4)–mAQP2 complex formed in kidney collecting ductal cells has been involved in the deubiquitinylation of AQP2 [75]. Data suggest that USP4 plays dual roles by limiting the extent of AQP2 endocytosis and by stabilizing AQP2 [75]. AQP2 ubiquitination by the NEDD4 and NEDD4L E3 ubiquitin ligases likely depends on the interaction between NEDD4 family-interacting proteins 1 and 2 (NDFIP1 and -2). NDFIP1/2 directly bind to the AQP2 C-terminus and this interaction is believed to recruit NEDD4 and NEDD4L, leading to ubiquitination and subsequent degradation [76]. The lysosomal degradation of AQP2 is also facilitated by the interaction with lysosomal trafficking regulator-interacting protein 5 (LIP5) [77], which binds the AQP2 C-terminal helix in a phosphorylation-dependent manner, with phosphorylation outside the binding site allosterically controlling the interaction [25,78]. Fluorescent spectroscopy revealed that the hAQP2–LIP5 interaction involves the LIP5 N-terminal domain and up to two LIP5 molecules per AQP2 tetramer (possibly dimerized through their C-terminal domain) despite the presence of four possible LIP5-binding sites. Furthermore, computational docking and mutational studies proposed a model whereby a MIT-interacting motif (MIM) present within the AQP2 C-terminal helix binds the LIP5 MIT-domain in a similar manner as LIP5 binding to components of the ESCRT-III complex [79]. Myelin and lymphocyte-associated protein (MAL) preferentially associated with hAQP2 phosphorylated on the Ser256 residue, and increased its phosphorylation and its expression at the plasma membrane by reducing its internalization [80]. Clathrin, clathrin adaptor protein (AP2), dynamin [66] and calveolin-1 [81] have also been shown to bind to hAQP2 and may be involved in its cellular redistribution. Actin [82,83,84,85] and tropomyosin 5b (TM5b) [86] have been shown to bind to the C-terminus of hAQP2. Upon the AVP-driven phosphorylation of AQP2, AQP2 separated from G-actin and bound to F-actin and TM5b [85,86]. The sequestration of TM5b from F-actin induced F-actin depolymerization and promoted AQP2 trafficking to the plasma membrane [86,87]. The N-terminal FERM domain of ezrin, an actin-binding protein from the ERM family, interacted with the C-terminus of AQP2 and promoted its endocytosis, thereby linking the actin skeleton dynamics with AQP2 trafficking [88].

Several proteins involved in calcium binding and handling have been identified to form a complex with AQP2. Annexin II, a multifunctional calcium- and lipid-binding protein, has been suggested to play a role in AQP2 trafficking [89] and was shown to interact with the C-terminus of AQP2 in a manner that was dependent on Ser256 phosphorylation [67]. In addition, both Transient Receptor Potential Cation Channel Subfamily V Member 4 (TRPV4) and small-conductance potassium channel (SK3) have been shown to bind to AQP2 [90]. AQP2 plays a key role in the activation of SK3 by TRPV4, leading to the hyperpolarization of the kidney collecting ductal cell plasma membrane that modulates the store-operated calcium entry (SOCE) [90]. The latter was shown to be involved in the AVP-induced translocation of AQP2 to the collecting duct apical plasma membrane [91].

In addition to this wide range of interacting proteins, AQP2 has been shown to interact with another AQP. Specifically, AQP2 was found to bind AQP5 in patients with diabetic nephropathy, but not in healthy controls. This interaction may participate in polyuria by impairing AQP2 localization to the plasma membrane [92]. However, the molecular basis of such an interaction remains to be assessed.

Overall, more studies will be required to fully understand the molecular basis of most of the protein–protein interactions involving AQP2, as well as their functional role in the complex AQP2 regulatory mechanism that underlies urine volume regulation.

## 5. AQP3

AQP3 is an aquaglyceroporin expressed in various sites within the body, including respiratory and digestive tracts, kidney, skin, and adipose tissue [93,94]. AQP3 is involved in upper or lower airway humidification [95,96], intestinal water reabsorption and the maintenance of intestinal epithelial barrier function [97] and urine concentration by the kidneys [98]. In addition, in the skin, AQP3 plays a role in cell migration and proliferation, epidermal water permeability barrier and skin hydration [99,100]. In adipose tissue, AQP3 could contribute to glycerol permeability and the exit of glycerol during lipolysis [94,101].

AQP3 has been shown to interact with the ClC-3 chloride channel in nasopharyngeal carcinoma, and this interaction has been described as playing an important role in cell volume regulation, involving water and glycerol fluxes, as well as in the opening of the ClC-3 channel [102,103]. Concomitant to chloride entry, hyperpolarization inhibited cell proliferation and promoted cell differentiation [104]. In keratinocytes, protein–protein interactions involving AQP3 are believed to play pivotal roles in cell differentiation and any perturbation may account for skin disease. Specifically, phospholipase D2 (PLD2) interacted with mAQP3 in caveolin-rich membrane microdomains, but no direct interaction between Caveolin-1 and AQP3 or PLD2 could be detected [105]. It was proposed that the PDL2–AQP3 interaction is involved in keratinocyte proliferation and differentiation, a hypothesis that is supported by several experimental studies. First, in keratinocytes, glycerol induced both growth arrest and differentiation; second, PLD2 metabolized phospholipids in the presence of glycerol to yield phosphatidylglycerol. The latter inhibited the proliferation, whereas the former induced the differentiation of keratinocytes [105] (Figure 5A). In contrast, a direct interaction between hAQP3 and the Caveolin-1 complex could be detected in extravillous trophoblast cells and functional data suggest that an altered AQP3–Caveolin-1 interaction may result in pregnancy disorders [106]. In silico analysis suggested that a putative calveolin-1 binding site located within the region between the third and fourth transmembrane domains (loop C) may be involved in the protein–protein interaction [106]. However, this hypothesis must be validated by additional experiments. Furthermore, it remains to be confirmed whether the AQP3–Caveolin-1 interaction occurs in keratinocytes.

The C-terminal part of perilipin-1 was shown to be involved in the interaction with hAQP3 [107] in addition to hAQP7 in adipocytes [107,108]. While AQP7–perilipin-1 binding was reduced upon the phosphorylation of a PKA site located within the N-terminus and triggered AQP7 translocation to the plasma membrane [108], the absence of a PKA site in AQP3 suggests that a distinct signaling pathway may control AQP3 [107] (Figure 5B). Further studies are required to identify this other signaling pathway and its role in the regulation of the AQP3–perilipin-1 complex and AQP3 functionality. Interestingly, in mouse 3T3-L1 adipocytes stimulated with lipopolysaccharides (LPS) that mimic inflammation occurring during obesity, the expression of both AQP3 and AQP7 is decreased through c-Jun N-terminal kinase (JNK)/NFkB signaling pathways and the AQP3 and AQP7 glycerol permeabilities are reduced probably concomitantly to protein internalization [101]. It would be interesting to determine what happens to the AQP3–perilipin-1 complex in response to inflammation and diabetes. Overall, additional studies will be needed to shed light on the role of perilipin-1 on AQP3 functionality under physiological conditions and metabolic diseases, such as obesity. It should be noted, however, that the expression and function of AQP3 and AQP9 in human adipocytes is controversial. A recent single-cell analysis of human adipose tissue confirmed the expression of AQP7 in mature human adipocytes, but the expression of AQP3 was identified in a low number of preadipocytes and mature adipocytes, AQP9 expression was shown in visceral adipose tissue progenitors, and AQP10 expression could not be detected [109]. In addition, the immunohistochemical labeling of AQPs in human adipose tissue suggests no functional overlap between AQP3/AQP9/AQP10 and AQP7 in human or mouse white visceral adipose tissue [109], as opposed to mouse 3T3-L1 adipocytes [101].

**Figure 5 ijms-23-09615-f005:**
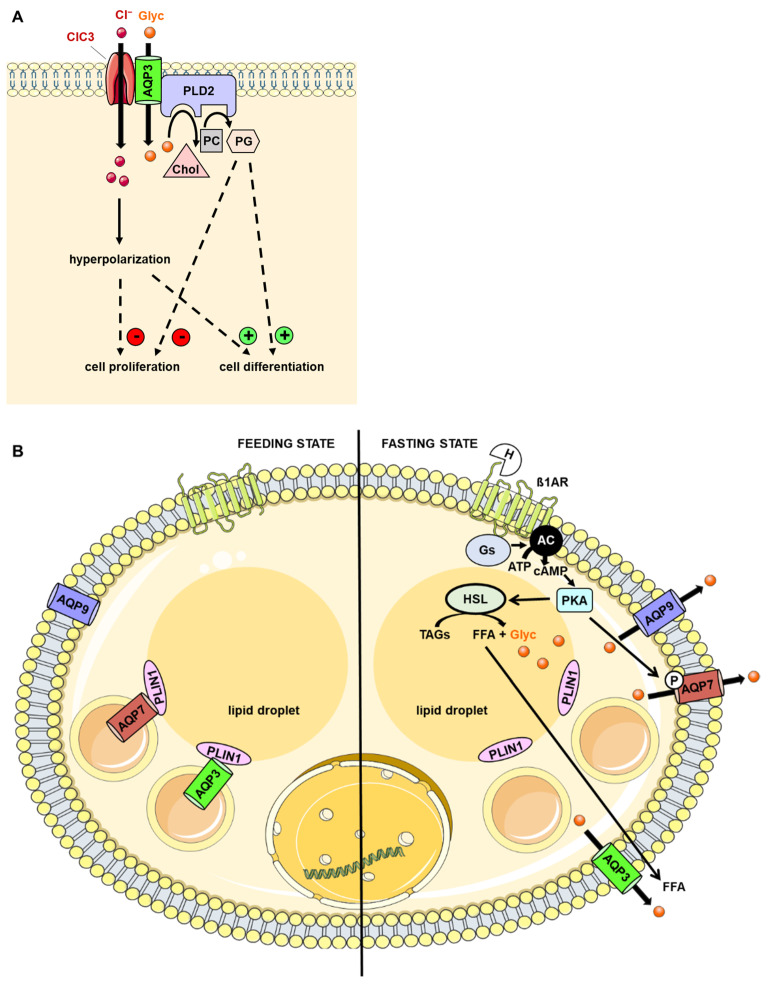
Role of the protein–protein interaction involving AQP3. (**A**) Role in keratinocyte differentiation. Upon glycerol entry through AQP3, glycerol is converted into PG by PLD2. PGly promotes growth arrest and differentiation of keratinocytes. (**B**) Role in adipocytes. PLIN1 interacts with AQP3 and AQP7 under feeding state. In fasting state, hormones, such as catecholamines, activate the ß-adrenergic receptor, which leads to a subsequent rise in cAMP and activation of PKA. PKA activates HSL hormone-sensitive lipase and thereby stimulates TAG degradation and Gly release. PKA also phosphorylates AQP7, which decreases it affinity for PLIN1 and enables its translocation to the plasma membrane. The AQP3–PLIN1 interaction is reduced by an unknow mechanism and leads to the translocation of AQP3 to the plasma membrane. AQP3 and AQP9 may only play a role in mouse 3T3-L1 adipocytes [101], but not in human or mouse white adipose tissue [109]. cAMP: cyclic adenosine monophosphate; Chol: choline; Glyc: glycerol; Gs: protein Gs; HSL: hormone-sensitive lipase; LIN-7: protein LIN-7 homolog; PC: phosphatidylcholine; PG: phosphatidylglycerol; PKA: protein kinase A; PLIN1: perilipin-1; PLD2: phospholipase D2; TAG: triacylglycerol.

## 6. AQP4

AQP4 is highly expressed within the central nervous system (CNS), kidney, eye, skeletal muscle, stomach, lung and other tissues [1,110]. Due to its involvement in CNS water homeostasis, AQP4 has emerged as a potential drug target in neurodegenerative diseases and CNS edema [111,112], and it was recently demonstrated that targeting AQP4 subcellular relocalization in astrocytes is a viable strategy for CNS edema following traumatic injury [111].

AQP4 exists in two splice variants, M1 and M23, that can assemble as both homo- and heterotetramers. The ratio between the M1 and M23 isoforms controls the formation of orthogonal array particles (OAP), supramolecular assemblies formed by AQP4 tetramers interacting with each other within the membrane, with a higher amount of AQP4–M23, leading to large OAPs and vice versa. Homotetramers of AQP4–M1 form no or very small OAPs and can move freely in the membrane, a property that has been proposed to promote cell migration [113]. In contrast, AQP4–M23 homotetramers form very large OAPs with low mobility, which is believed to help maintain the polarized expression of AQP4 in astrocyte end-feet by supporting the interaction between AQP4 and the extracellular matrix [114] and the cytoskeleton [115,116,117,118].

The role of AQP4 tetramerization was characterized by creating AQP4 loop D mutants that had different propensities to oligomerize. This study shows that AQP4 tetramerization is not a prerequisite for targeting the plasma membrane or water permeability. However, the ability to control plasma membrane abundance following tonicity changes was lost for loop D mutants that were unable to tetramerize, suggesting that tetramerization and/or interactions with other proteins via loop D are essential for tonicity-triggered changes in AQP4 subcellular localization [6]. Further studies will be needed to determine if tetramerization also plays a similar role for other AQPs.

In the brain, several proteins found to interact with AQP4 have some functional involvement (Figure 6). Transient Receptor Potential Vanilloid 4 (TRPV4) has been shown to interact with rAQP4, mAQP4 and hAQP4 and the AQP4–TRPV4 complex may play an essential role in the hypotonicity-induced increase in intracellular calcium and inhibition of regulatory volume decrease [119]. Indeed, based on experimental data, it has been postulated that AQP4-mediated water flux activates TRPV4 and, reciprocally, calcium entry through TRPV4 modulates AQP4-mediated water movement [120,121]. AQP4 has also been shown to interact with another member of the TRP family, TRPM4 (Transient Receptor Potential Cation Channel Subfamily M Member 4 (TRPM4), a non-selective monovalent cation channel activated by intracellular calcium [122]. Together with Sulfonylurea receptor 1 (SUR1), an ATP-binding cassette transporter regulating pore-forming units [123], AQP4 and TRPM4 may form a ternary complex that is involved in cell volume regulation [121,123]. A model of the SUR1–TRPM4–AQP4 complex has been proposed in which four TRPM4 subunits are surrounded by four SUR1 subunits, with four AQP4 tetramers intercalated in between [123,124], although this remains to be confirmed by structural studies. SUR1 binding to TRPM4 increased its intracellular calcium sensitivity, contributing to an augmentation of the role of TRPM4 as a negative regulator of intracellular calcium influx, an effect that was further amplified by water influx through AQP4, thereby further diluting the calcium concentration [124]. Studies have shown that TRPM4-knockout mice displayed altered cell volume regulation involving AQP4 [121,123], and the co-expression of TRPM4 and AQP4 did not result in higher water flow, unlike the co-expression of SUR1, TRPM4 and AQP4 [123]. Interestingly, CNS injury has been proposed to cause AQP4 to switch interaction partners, from TRPV4 under normal physiological conditions to SUR1–TRPM4, forming a water/ion channel complex that drives edema formation [123].

The interaction between AQP4 and CaM is important for AQP4 to target the plasma membrane [111,125] and blocking it using a CaM antagonist significantly improved recovery following traumatic CNS injury in rats [111]. AQP4 binds CaM in a calcium-dependent manner (Kd: 29 ± 6 µM) and, based on sequence analysis, the CaM binding site was proposed to be located at residues 256–275. While there is no experimental structural information for this region, it is predicted to form an amphipathic helix, as seen in other mammalian AQPs, with three phenylalanine residues forming a hydrophobic CaM-binding motif. This hypothesis was supported by the finding that replacing the three phenylalanine residues with alanine abolished binding [111]. A recent study using AQP4 N- and C-terminal peptides revealed CaM-binding domains located within both termini, with the N-terminus interacting with the N-lobe of calmodulin (Kd: 3.4 ± 0.8 µM, 1:1 binding ratio) and the C-terminus interacting with both the C- and N-lobes of CaM (Kd1: 3.6 ± 1.3 µM; Kd2: 113.6 ± 8.0 µM, 2:1 binding ratio). Moreover, both peptides could interact with CaM simultaneously, forming a ternary complex [126]; however, it remains to be established if this is also possible in the full-length context. It has been debated whether the AQP4–CaM interaction is strengthened [111] or not [126] upon AQP4 phosphorylation at Ser276, a PKA site that has been shown to be involved in AQP4 plasma membrane localization [111]. As this apparent discrepancy might be related to the use of different methodologies, further studies will be required to clarify this.

AQP4 has been demonstrated to interact with the dystrophin–glycoprotein complex (DGC), a multimeric transmembrane protein complex that was first isolated from muscle tissue but is also present non-muscle tissues, including the CNS [127]. The interaction between AQP4 and DGC is mediated by one of its components, α-syntrophin, via a PDZ-domain located in the AQP4 C-terminus. It has been hypothesized that this helps anchor AQP4 in the astrocyte end-feet plasma membrane, thereby being important for the polarized expression of AQP4 in these cells [115]. Indeed, α-syntrophin-knockout mice displayed reversed AQP4 expression in astrocytes as well as an overall decrease in expression in muscle [115]. Moreover, there was an impaired water flux and potassium buffering concomitantly to an AQP4 localization away from glial end-feet [128,129]. The effect on potassium is believed to arise from a modulation of the inwardly rectifying potassium channel Kir4.1 and it was shown that the ischemia-induced loss of Kir4.1 increased glial cell swelling under hypotonic conditions [130]. Since Kir4.1 has also been shown to interact with the DGC, the formation of an AQP4–Kir4.1–DGC multiprotein complex has been hypothesized. However, the existence of such a complex and the ability of AQP4 to modulate Kir4.1 activity is a subject of controversy [131,132].

The glutamate transporter 1 (GLT1) and the mu-opioid receptor (MOR) interacted with rAQP4 [133], but their impact on AQP4 functionality remains unknown. GLT1, a co-transporter for sodium and glutamate, interacted with the AQP4 C-terminus (residues 252–323) [133]. Furthermore, the Na,K-ATPase and the Metabotropic glutamate receptor 5 (mGluR5) have both been shown to interact with the N-terminus of rAQP4 (residues 23–32) and regulated its water permeability [134]. The interactions between Na,K-ATPase, mGluR5, GLT1and rAQP5 are likely involved in neuron–astrocyte crosstalk under physiological and pathological conditions [133,134]. In addition, the cystic fibrosis transmembrane conductance regulator (CFTR) binds AQP4 in Sertoli cells [135]; however, the physiological significance of this interaction remains to be determined.

The µ-subunit of clathrin assembly protein complexes 2 and 3 (µAP2 and µAP3) have been shown to bind to rAQP4 and play a role in its trafficking [136], with AP2 and AP3 being involved in endocytosis and lysosomal targeting, respectively. The interaction was proposed to be mediated by an endocytic motif in the AQP4 C-terminus (GSYMEV), with Y277 being critical for the interaction. Moreover, the casein kinase II-induced phosphorylation of Ser276 increased the amount of AQP4–µAP3 interactions, whereas it had no effect on the interaction between AQP4 and µAP2 [136]. These data suggest that protein–protein interactions modulating AQP4 lysosomal targeting steps may control plasma membrane water permeability, thereby playing an important role in cerebral water balance.

These compelling observations suggest that protein–protein complexes involving AQP4 play fundamental roles in the physiological and pathological processes of the astrocytes in the CNS. In this pathophysiological context, the use of inhibitors targeting interacting partners that control AQP4 localization and function, in addition to inhibitors targeting AQP4 itself, may represent a promising therapeutic prospect [111,137,138].

## 7. AQP5

AQP5 is predominantly expressed in secretory glands, such as salivary, sweat, lacrimal, and mammary glands, and in lungs and airways, where it plays a central role in the production of saliva, sweat, tears and pulmonary secretions [96,139,140,141]. More recently, AQP5 has been studied in the ocular lens [142,143].

To date, the mechanisms regulating AQP5 trafficking from intracellular structures to the plasma membrane still remain poorly understood. AQP5 movement to the plasma membrane is triggered upon acetylcholine (ACh) and noradrenalin (NA) binding to M3 and β adrenergic receptors [144,145], which results in a rise in cytoplasmic calcium and cAMP concentrations and the subsequent activation of protein kinases. Although two consensus phosphorylation sites (Ser156 and Thr259) are present in cytoplasmic loop D and the C-terminus of AQP5, the correlation between their phosphorylation and trafficking signals have not been fully determined [125,146]. It has been proposed that at least three independent mechanisms may account for AQP5 trafficking: Ser156 phosphorylation, protein kinase A activation, and extracellular osmolarity [125]. In the last few years, key roles have been attributed to AQP5-interacting protein partners that can coordinate and regulate its trafficking and function in mammalian secretory cells; nevertheless, the AQP5 interactome remains largely unexplored. Only a few AQP5-interacting protein partners have been identified, such as Na-K-Cl cotransporter 1 (NKCC1), anion exchanger 2 (AE2) [147], TRPV4 [148], Mucin 5AC (MUC5AC) [149], prolactin-inducible protein (PIP) [150,151] and ezrin [152] (Figure 7); however, the molecular basis for these interactions remains to be fully elucidated.

The ability of salivary gland cells to avoid osmotic imbalance is prevalently due to the movement of water through the AQP5 channel after solute efflux via ion transporters, such as NKCC1 and AE2, wherefore crosstalk and direct or indirect interactions between ion transporters and AQP5 may guarantee volume regulation. NKCC1 and AQP5 colocalization was observed at the lumen of mouse salivary gland acinar cells and in HEK293T overexpressed system. The AQP5–AE2 protein–protein interaction was also established in submandibular gland cells and HEK293T overexpression system [147]. Additional experiments are necessary to uncover the underlying structural interactions between these proteins.

The functional and physical relation between osmosensing TRPV4 and AQP5 was identified in salivary glands. The expression of N-terminus-deleted AQP5 in salivary gland cell lines had a dominant negative effect, reducing the formation of functional AQP5 channels and suppressed TRPV4 activation and regulatory volume increase (RVD). These data suggest that the activation and regulation of TRPV4 function by hypotonicity depends on AQP5 interactions [148].

While AQP5 was shown to interact through its C-terminal with PIP in normal mouse lacrimal gland, this interaction was lost in mice with Sjögren’s syndrome [150]. More recently, the AQP5–PIP interaction has also been shown in human salivary glands and a human salivary gland cell line, and a deeper understanding of the AQP5–PIP interaction has been provided [151]. Indeed, it has been shown that the PIP can bind the AQP5 C-terminus with a Kd value of 0.57 ± 0.07 μM and with a ratio of one molecule of PIP per AQP5 tetramer. These interactions did not involve the AQP5 C-terminal helix as the truncated construct (T242Stop) that retains this region lost its affinity for PIP, suggesting that the PIP binding site may be located in the distal part of the C-terminus [151]. This hypothesis was supported by a previous study in lacrimal glands showing the recovery of PIP following pulldown assay using a C-terminal peptide corresponding to residues 251–265 [150]. PIP may act as an adaptor between AQP5 and the actin cytoskeleton during AQP5 trafficking. Indeed, reduced PIP expression in salivary glands from Sjögren’s Syndrome (SS) patients resulted in abnormal AQP5 localization and trafficking to the apical membrane of acinar cells [151].

The interaction between AQP5 and ezrin was first observed in human and mice salivary glands [152]. The physical interaction of AQP5 and ezrin was corroborated in eukaryotic cells and in healthy and pathologic human salivary glands tissues by proximity ligation assay and immunoprecipitation. In the proposed docking model of this complex, AQP5 interacts with the ezrin FERM domain via its C-terminus, specifically via residues 247–263, a region that is absent in the crystal structure but is predicted to form an alpha helix. Based on this model and sequence analysis, a novel consensus motif for how FERM domains interact with helical peptides was proposed [152], corresponding to a modification of the previously proposed motif for the interaction between NHERF and the radixin/moesin FERM domains [153]. Furthermore, the docking model of the AQP5-FERM domain complex showed that the helical segment is able to bind the FERM domain in the opposite direction [152]. Further studies are required to clarify whether this observation is a docking artefact or a true flexibility in the binding mode. As observed for PIP, the reduced ezrin expression in salivary glands from Sjögren’s syndrome patients resulted in abnormal AQP5 localization [152] and trafficking [154]. Altogether, these data suggest that both PIP and ezrin could be part of an intricate network of proteins likely regulates AQP5 function and localization.

An AQP5–MUC5AC complex was observed in the conjunctiva of rabbit as well as in mouse dry eye models. Specifically, immunoblotting against AQP5 and MUC5AC in the total protein homogenates isolated from the rabbit conjunctiva showed a positive and synchronous expression pattern with progressive upregulation in dry eye mice models. In addition, immunoprecipitation with anti-AQP5 or MUC5AC antibodies confirmed the AQP5–MUC5AC interaction. This complex appears to be critical to ensure an adequately hydrated mucus gel, and to maintain normal properties of the eye [149]. It remains to be determined whether this the complex between AQP5 and MUC5AC also exists in salivary glands.

In lens fiber cells, AQP5 plasma membrane insertion occurs over time in contrast to the immediate insertion of AQP0 [143]. Interestingly, the colocalization of AQP5 with mitochondrial and autophagic markers as well as the inhibition of trafficking via autophagosome/lysosome fusion inhibitor, bafilomycin A1, suggests a novel mechanism of AQP5 trafficking, namely the unconventional protein secretion (UPS) process of lysosomal secretion [155]. In addition, AQP5 insertion into lens fiber cell membranes can be regulated by mechanical tension and pharmacologic treatment with a muscarinic agonist, pilocarpine [156]. Direct AQP5–protein interactions involved in these processes remain to be identified.

## 8. AQP6

AQP6 is primarily found in intracellular vesicles within acid-secreting alpha-intercalated cells from the renal collecting ducts [157]. AQP6 has low water permeability and works as an pH-dependent anion channel [158] that may be important for urinary acid secretion [159].

CaM has been shown to interact with the N-terminus of AQP6 from mice, rats and humans in a calcium-dependent manner, with an apparent Kd of 0.79 ± 0.08 µM and 1.78 ± 0.13 µM for mAQP6 and hAQP6 N-terminal peptides, respectively [159]. The predicted binding site contains a canonical CaM-binding motif, which was confirmed by mutational studies. As the N-terminus of AQP6 is necessary for protein trafficking to the intracellular vesicles [160], CaM may play a role in this process. To date, the role of CaM in AQP6 function remains to be elucidated.

## 9. AQP7

AQP7 is an aquaglyceroporin expressed in many tissues and organs including adipose tissue, kidney and pancreas [93].

In adipocytes, AQP7 interacted with perilipin-1 (PLIN1), a highly abundant protein expressed at the lipid droplet surface where it is involved in triacylglycerol mobilization [108]. The C-terminal part of PLIN1 interacted with AQP7 N-terminus [107,108]. Moreover, the AQP7–PLIN1 complex was dissociated following stimulation by catecholamines (fasting conditions), allowing AQP7 to translocate to the plasma membrane to ensure glycerol efflux. In contrast, the protein complex was retained at the lipid droplet surface following administration of insulin (feeding condition). Since catecholamines and insulin stimulate and inhibit PKA, respectively [161], it was suggested that the AQP7–PLIN1 interaction may play a key role in triacylglycerol metabolism under physiological conditions as well as in metabolic diseases such as obesity (Figure 5B). Further studies will be required to confirm this hypothesis.

## 10. AQP8

AQP8 is expressed in various cells and tissues, including spermatozoids, liver, salivary glands [162,163,164], and has been found in both the plasma membrane and the inner mitochondrial membrane [165,166]. Similarly to some AQPs (AQP0 [167], AQP1 [168], AQP3 [169], AQP5 [170], AQP9 [171], and AQP11 [172]), AQP8 is considered as a peroxiporin due to its permeability to H_2_O_2_ in addition to water [173,174,175]. Moreover, several studies have shown that AQP8 is also highly permeable to ammonia, which may play an important role in nitrogen metabolism [176,177].

Human Papilloma Virus L1 protein (HPV L1) binds to hAQP8 and reduces AQP8 water permeability in spermatozoids [178]. Further studies are necessary to identify which part of AQP8 is involved in this interaction and to determine whether the HPV L1–AQP8 interaction may also reduce the elimination of H_2_O_2_ from the spermatozoids or their mitochondria.

## 11. AQP9

AQP9 is an aquaglyceroporin that is expressed in various tissues including liver, testes, eye, and intestine [93,94]. AQP9 is the most abundant AQP expressed in male reproductive tract. During spermatogenesis, the volume modification of differentiating germ cells is predominantly due to the osmotically driven fluid efflux through AQPs.

The interaction between AQP9 and monocarboxylate transporters 1, 2 and 4 in mouse retina has been hypothesized to contribute to the transport of lactate as an energy substrate by participating in the astrocyte-to-neuron lactate shuttle (ANLS), thereby playing a key function in neuron metabolism [179]. Noteworthy, AQP9 has also been shown to be permeable to monocarboxylates such as lactate [180], suggesting a direct contribution to the ANLS. The roles of such AQP9–protein interactions are summarized in Figure 8A. Additional studies are required to deepen our understanding of the molecular basis and the role of this protein–protein interaction in physiological and pathophysiological conditions.

The interaction between cystic fibrosis transmembrane conductance regulator (CFTR) and rat AQP9 in testes Sertoli cells may be of physiological importance as CFTR is proposed to act as a regulator of AQP9 permeability involved in cell volume regulation during spermatogenesis [181,182]. This hypothesis was corroborated by experiments showing that CFTR potentiated AQP9 water and glycerol permeabilities in *Xenopus* oocytes and perfused rat epididymis [182,183]. While it has been shown that CFTR interacted with AQP9 through its first nucleotide-binding domain (NBD1) [183], further studies are warranted to assess the AQP9 domain involved in this protein–protein interaction. The PDZ1 and PDZ2 (PSD-95, Drosophila disc large protein, ZO-1) domains of the Na+/H+ Exchanger Regulatory Factor (NHERF1), know to regulate CFTR, interact with the C-terminus of AQP9 which contains a putative PDZ binding motif (SVIM) [182]. A functional complex involving AQP9, CFTR and NHERF1 may take part in water and solute transport in the male reproductive tract and its disruption could contribute to the pathogenesis of male infertility in cystic fibrosis [184] (Figure 8B).

## 12. Conclusions

The mammalian AQPs’ interactome is highly complex and new interaction partners are continuously being discovered. Table 1 provides a summary of the interacting proteins for which the interactions have been best characterized. It is becoming increasingly clear that these protein–protein interactions play crucial roles in controlling and modulating AQP function. A deeper understanding of these regulatory processes, including at the molecular level, will be key to fully understand the tissue-dependent regulation of AQPs and their role in health and disease. In the future, sequence-based predictions of protein–protein interactions, bioinformatic compilation of the protein–protein interactome, spatial interactome, live-imaging of protein–protein interactions, biophysical and biochemical methods, new photoproximity protein profiling interaction methods, mass spectrometry-based proteomic, and thermal proximity coaggregation profiling are anticipated to further uncover the spatial and temporal AQPs’ interactome that will contribute to a more in-depth understanding of physiological and pathophysiological molecular mechanisms. Moreover, the structures of AQP complexes of medical relevance are highly interesting for drug design purposes. Here, the development and resolution revolution of single-particle cryo-electron microscopy for membrane protein structural studies is likely to play an important role [185]. Given that a structure-based drug design targeting individual AQPs remains challenging despite intense efforts [186,187,188,189,190], switching the focus to the AQP interactome may offer new therapeutics avenues for the treatment of human disease states in which AQPs play an important role.

## Figures and Tables

**Figure 1 ijms-23-09615-f001:**
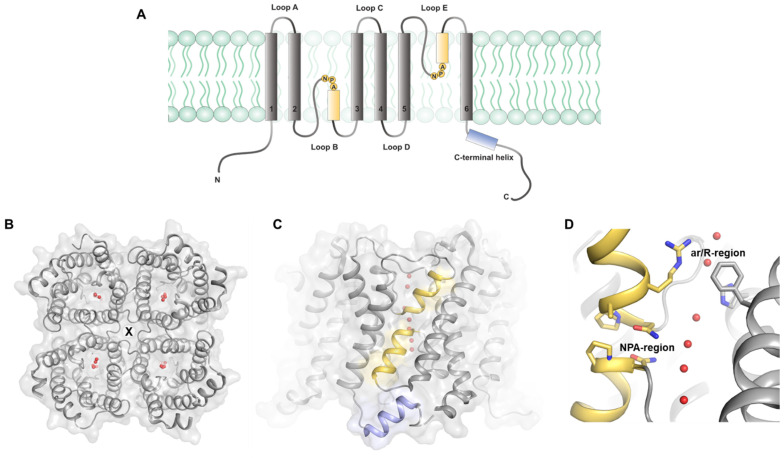
Structural features of mammalian AQPs. (**A**) Schematic representation of the mammalian AQP topology. Each monomer comprises six transmembrane helices (I–IV) connected by five loops (A–E). Loops B and E fold back into the membrane, forming two half-membrane spanning helices (highlighted in yellow) that harbor the two asparagine–proline–alanine (NPA) motifs. The N- and C-termini are located in the cytoplasm, with the C-terminus typically forming a short amphipathic helix (highlighted in blue) that is a common protein–protein interaction site. (**B**) Crystal structure of the AQP5 tetramer viewed from the cytoplasmic side (PDB code 3D9S), showing the four individual water channels formed by the monomers with water molecules drawn as red spheres. The central channel through the middle of the tetramer is indicated with an X. (**C**) Crystal structure of the AQP5-tetramer viewed from the side of the membrane. The loop B and E half-membrane spanning helices and the C-terminal helix are colored yellow and blue, respectively. (**D**) Zoom-in on the water-conducting channel where water molecules line up in a single file. Residues in the aromatic–arginine (ar/R) and asparagine–proline–alanine (NPA) regions are shown in stick representation. The two half-membrane spanning helices are colored yellow.

**Figure 2 ijms-23-09615-f002:**
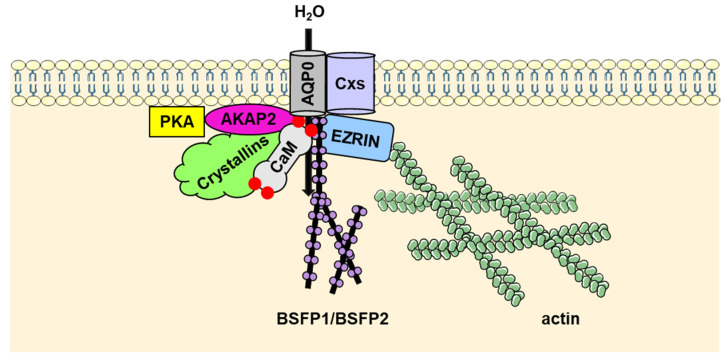
Roles of the protein–protein interactions involving AQP0. In the developing lens fiber cell, AQP0 and Cxs join to modulate cell–cell adhesion. AQP0 interacts with ezrin to directly link the plasma membrane with the actin cytoskeleton. Lens-specific beaded intermediate filaments, BFSP1 and BFSP2, also interact with AQP0 to link the plasma membrane to cytoskeletal structures. BSFP1 fragments can reduce AQP0 water permeability. AQP0 interacts with CaM to regulate water permeability. Interactions with AKAP2 modulate AQP0 phosphorylation, which affects CaM binding and water permeability. Interactions with abundant crystallin proteins occur, but with unknown functional consequences. AKAP2: A-kinase anchoring protein 2; BFSP1: beaded filament structural protein 1 (also called filensin); BFSP2: beaded filament structural protein 2 (also called 49 kDa Cytoskeletal Protein (CP49) or phakinin); CaM: calmodulin; Cxs: connexins; PKA: protein kinase A.

**Figure 3 ijms-23-09615-f003:**
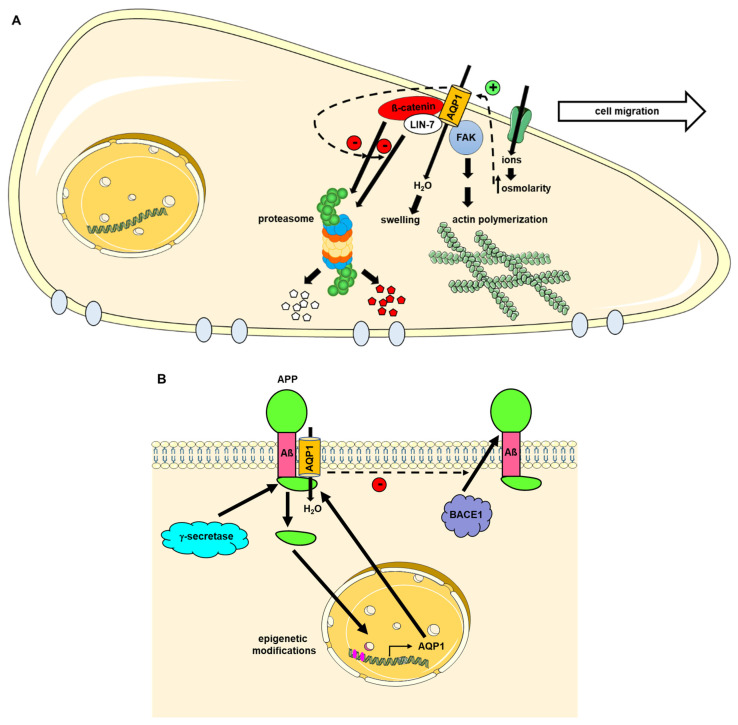
Role of the protein–protein interactions involving AQP1. (**A**) Role in cell migration. During cell migration, increased ion transport at the leading edge of the cells raises intracellular osmolarity that subsequently drives water entry through AQP1. AQP1 binding to FAK, LIN-7 and ß-catenin promotes actin polymerization, at least partly by inhibiting LIN-7 and ß-catenin degradation through the proteasome. (**B**) Role in APP accumulation in Alzheimer’s disease. AQP1 interaction with APP reduces the binding of BACE1 to APP and inhibits the release of Aß. In addition, the N-terminus of APP released by γ-secretase induces epigenetic modification, leading to increased AQP1 expression. Aß: amyloid-ß peptide; APP: amyloid precursor protein; BACE1: ß-secretase; FAK: Focal-adhesion kinase; LIN-7: protein LIN-7 homolog.

**Figure 4 ijms-23-09615-f004:**
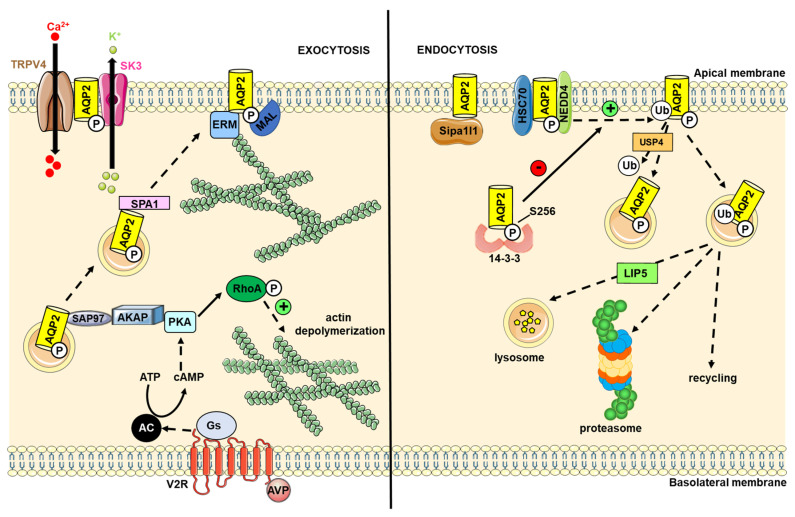
Role of the protein–protein interaction involving AQP2 in kidney collecting duct cells. Upon AVP stimulation, leading to subsequent cAMP increase and PKA activation, AQP2 moves to the plasma membrane thanks to the help of various protein partners. In the absence of hormonal stimulation, AQP2 endocytosis also requires many protein partners. Due to the high number of proteins that bind to AQP2, only some of them are indicated in this figure. 14-3-3: protein 14-3-3-; AC: adenylyl cyclase; AKAP: A-kinase-anchoring protein; AVP: arginine vasopressin; cAMP: cyclic adenosine monophosphate; ERM: ezrin/radixin/moesin protein family; Gs: protein Gs; HSC70: heat shock protein 71 kDa protein; LIP5: lysosomal trafficking regulator-interacting protein 5; MAL: myelin and lymphocyte-associated protein; NEDD4: E3 ubiquitin protein ligase NEDD4; PKA: protein kinase A; PKC: protein kinase C; RhoA: small GTP-Binding Protein RhoA; SAP97: synapse-associated protein 97; Sipa1l1: signal-induced proliferation-associated 1 like 1; SPA1: signal-induced proliferation-associated gene-1; SK3: small-conductance potassium channel; TRPV4: Transient Receptor Potential Cation Channel Subfamily V Member 4: USP4: ubiquitin-specific peptidase 4.

**Figure 6 ijms-23-09615-f006:**
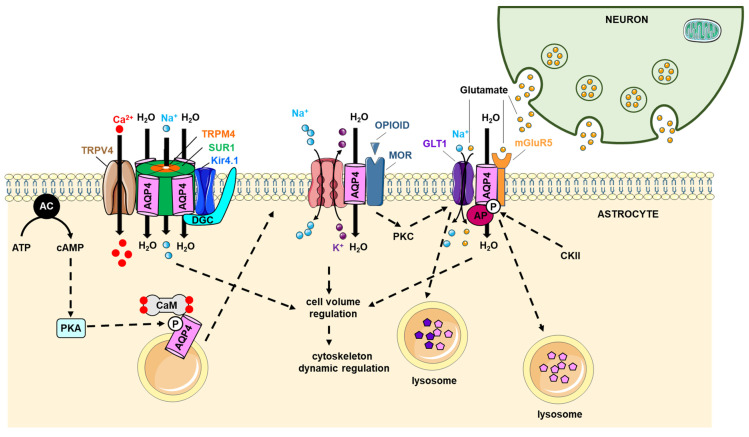
Role of the protein–protein interaction involving AQP4 in astrocytes. The AQP4–TRPV4 interaction plays a role in AQP4 trafficking for calcium entry via TRPV4 binding to CaM. CaM binding, reinforced by AQP4 phosphorylation by PKA, enables AQP4 to move to the plasma membrane. In the plasma membrane, AQP4 also interacts with SUR1, TRPM4, Kir4.1, and DGC. AQP4 also binds to MOR, GLT1, AP, Kir4.1 and Na,K ATPase. CKII can phosphorylate AQP4 to promote its lysosomal targeting and degradation. These protein–protein interactions are likely playing a role in cell volume regulation and cytoskeleton dynamics. AP: clathrin assembly protein complex; CaM: calmodulin; CKII: casein kinase II; DGC: dystrophin-glycoprotein complex; GLT1: glutamate transporter 1; Gs: protein Gs; Kir4.1: inwardly rectifying potassium channel Kir4.1; mGluR5: metabotropic glutamate receptor 5; MOR: mu opioid receptor; Na,K ATPase: sodium, potassium ATPase pump; PKA: protein kinase A; PKC: protein kinase C; SUR1: Sulfonylurea receptor 1; TM2: transmembrane domain 2; TRPM4: Transient Receptor Potential Cation Channel Subfamily M Member 4; TRPV4: Transient Receptor Potential Cation Channel Subfamily V Member 4; µAP 2/3: clathrin assembly protein complexes 2 and 3 µ subunit.

**Figure 7 ijms-23-09615-f007:**
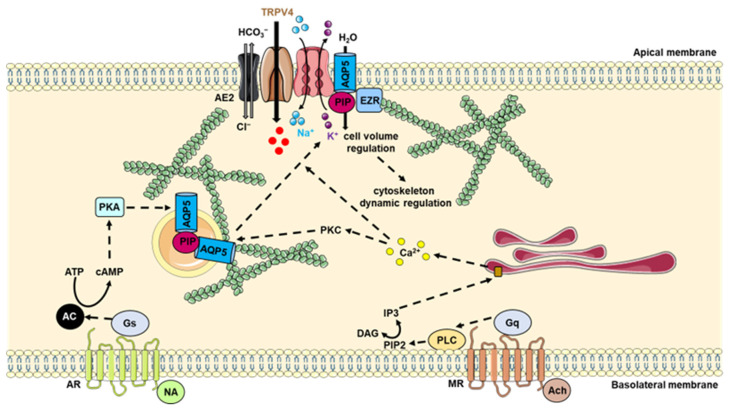
Role of the protein–protein interaction involving AQP5 in salivary gland epithelial cells. Nervous stimulation induces the release of both NA and Ach, which bind to AR and MR. AR activation induces the subsequent activation of Gs protein and AC. AC induces an increase in intracellular levels of cAMP, which activates PKA. MR activation induces the subsequent activation of protein Gq and PLC. PLC cleaves PIP2 into IP3 and DAG. IP3 stimulates calcium release from endoplasmic reticulum and the activation of PKC. These stimuli induce AQP5 trafficking to the plasma membrane. The interaction between AQP5 and NKCC1, AE2 and TRPV4 is likely involved in cell volume regulation and cytoskeleton dynamic regulation, while the interaction between AQP5 and PIP and EZR is likely involved in AQP5 trafficking to the apical plasma membrane. AC: adenylyl cyclase; Ach: acetylcholine; AE2: anion exchanger 2; ß1AR: β1 adrenergic receptors; cAMP: cyclic adenosine monophosphate; DAG: diacylglycerol; EZR: ezrin; Gq: protein Gq; Gs: protein Gs; H: hormone; IP3: inositol 1,4,5 triphosphate; MR: M1 and M3 subtypes of muscarinic receptors; NA: noradrenaline; NKCC1: Na-K-Cl cotransporter 1; PIP: prolactin-inducible protein; PIP2: phosphatidylinositol 4,5-bisphosphate; PLC: phospholipase C; PKA: protein kinase A; PKC: protein kinase C; TRPV4: Transient Receptor Potential Cation Channel Subfamily V Member 4.

**Figure 8 ijms-23-09615-f008:**
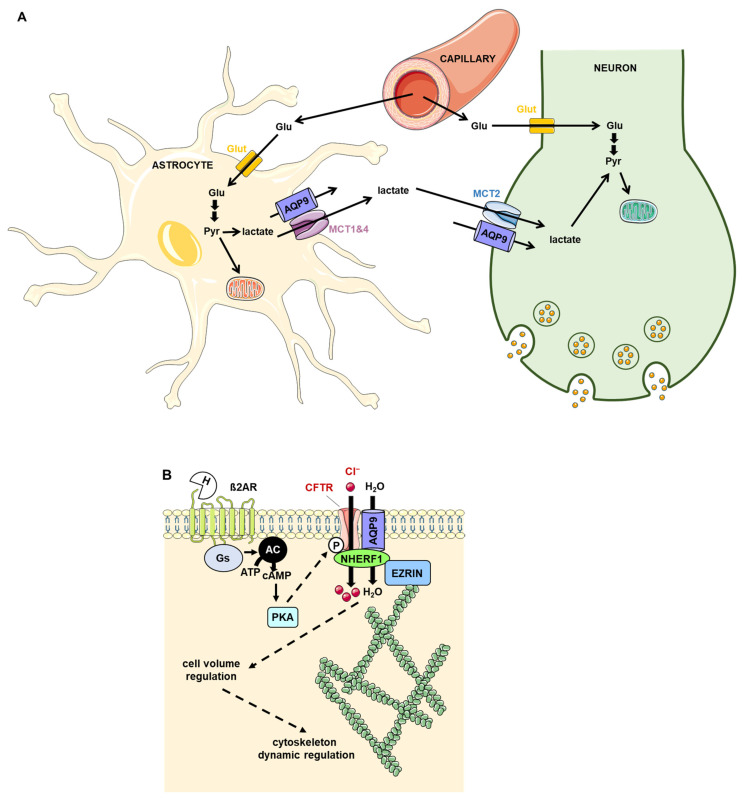
Role of the protein–protein interaction involving AQP9. (**A**) Role in the astrocyte-to-neuron lactate shuttle. Glu enters astrocytes and neurons via GLUT and is transformed into Pyr. In astrocytes, Pyr either used by the mitochondria to produce ATP or is converted to lactate which exits the astrocytes through MCTs 1 and 4. Then, lactate enters neurons via MCT2 and is converted into Pyr, which is used to produce ATP in the mitochondria. Lactate may also directly pass through AQP9. (**B**) Role in vas deferens cells. Upon adrenergic stimulation, ß2AR is activated and leads to subsequent PKA activation, which in turn phosphorylates CFTR. CTRF–AQP9, CTRF–NHERF1 and NHERF1–AQP9 are likely involved in cell volume control. In addition, the NHERF1–Ezrin interaction is likely coupling cell volume control to cytoskeleton dynamics. AC: adenylyl cyclase; ß2AR: ß2 adrenergic receptor; CFTR: cystic fibrosis transmembrane conductance regulator; Glu: glucose; GLUT: glucose transporter; Gs: protein Gs; H: hormone; MCT: monocarboxylate transporters; NHERF1: Na^+^/H^+^ Exchanger Regulatory Factor; PKA: protein kinase A; Pyr: pyruvate.

**Table 1 ijms-23-09615-t001:** AQP interacting partners.

AQP	Interacting Partner	AQP Region Involved in PPI	Detection Method
AQP0	CaM	C-term [27,191]	CL-EM [23]; MST [25]; CL-MS [27]
	ezrin	C-term [31]	CL-MS [31]
	BFSP1 (filensin)	C-term [35,36]	AFC [35] CL-MS [36]
	BFSP2 (CP49)	C-term [35]	AFC [35]
	connexins	C-term [40]	IP [40]
	AKAP2	C-term [28]	coIP [28]
	crystallins	C-term [46]	IP, YTH [46]; CL-MS [47]
AQP1	Lin7	ND	IP [48]
	FAK	ND	IP [49]
	ß-catenin	ND	IP [49]
	APP	N-term [56]	IP, OVL [56]
AQP2	ezrin	C-term [88]	IP [88]
	Hsc70	C-term [66]	IP, PDA [66,67]
	Hsp70	C-term [68]	IP [67,68]
	Clathrin	ND	IP [66]
	AP2	ND	IP [66]
	Dynamin	ND	IP [66]
	Sipa1l1	C-term [69]	IP [69]
	14-3-3 θ/ζ	C-term [74]	IP [74]
	NEDD4	ND	IP [76]
	NEDFIP 1/2	ND	IP [76]
	USP4	ND	IP [75]
	LIP5	C-term [78]	FS [79], MST [25,78], YTH [77]
	SNX27	C-term [192]	IP, PDA [192]
	TRPV4	ND	IP [90]
	SK3	ND	IP [90]
	Caveolin-1	ND	IP [81]
	MAL	C-term [80]	IP [80]
	Spa1	ND	IP [70]
	AKAP220	ND	YTH [72]
	Annexin II	C-term [67]	IP [67]
	actin	C-term [85]	IP [82,84]
	TM5b	C-term [86]	IP [67]
	AQP5	ND	IP [92]
AQP3	ClC3	ND	IP [102]
	PDL2	ND	IP [105]
	Perilipin-1	ND	BN-PAGE [107]
	Caveolin-1	ND	IP [106]
AQP4	AQP4	IC loop D, TM2 bottom [6]	FRET, MDS [6]
	CaM	C-term [111,126]; N-term [126]	MST [111]; NMR [126]
	DGC	ND	IP [193,194,195]
	Kir4.1	ND	IP [193]
	TRPV4	ND	IP [119]
	TRPM4	ND	IP, FRET [123]
	SUR1	ND	IP, FRET [123]
	CFTR	ND	IP [135]
	GLT1	C-term	IP, FRET [133]
	MOR	ND	IP [133]
	Na,K-ATPase	N-term [134]	IP, FRET [134]
	mGluR5	N-term [134]	IP, FRET [134]
	µAP 2/3	C-term [136]	YTH [136]
AQP5	PIP	C-term [150,151]	PDA [150], PLA, MST [151]
	ezrin	C-term [152]	SILAC-IP-MS, PLA [152]
	NKCC1	ND	IP [147]
	TRPV4	N-term [148]	IP [148]
	Mucin 5AC	ND	IP [149]
	AE2	ND	IP [147]
AQP6	CaM	N-term [157]	PDA [157]
AQP7	Perilipin-1	N-term [108]	PDA, PLA [108]
AQP8	HPVL1	ND	IP [178]
AQP9	CFTR	ND	IP [181]
	NHERF1	C-term [182]	PDA, OVL [182]
	MCTs	ND	IP [179]
AQP10	ND	ND	-
AQP11	ND	ND	-
AQP12	ND	ND	-

AE2: anion exchanger 2; AFC: affinity chromatography; AKAP2: A-Kinase Anchoring Protein 2; APP: amyloid precursor protein; BFSP1: beaded filament structural protein 1 (also called filensin); BFSP2: beaded filament structural protein 2 (also called 49 kDa Cytoskeletal Protein (CP49) or phakinin); BiP: hsp70 isoform 5; BN-PAGE: blue native polyacrylamide gel electrophoresis; CaM: calmodulin; CFTR: cystic fibrosis transmembrane conductance regulator; CL-EM: cross-linking followed by electron microscopy; CL-MS: cross-linking mass spectrometry; C-term: C-terminus; ClC3: voltage-gated chloride channel 3; CoIP: co-immunoprecipitation; Cxs: connexins; DGC: dystrophyn-glycoprotein complex; FAK: focal adhesion kinase; FRET: Forter Resonant Energy Transfer; FS: fluorescent spectroscopy; GLT1: glutamate transporter 1; Hsc70: Heat shock cognate 71 kDa protein; HPV L1: human Papillomavirus L1 protein; HSP70: heat shock protein of 70kDa; IC: intracellular; IP: immunoprecipitation; Kir4.1: inwardly rectifying potassium channel Kir4.1; LIN-7: protein LIN-7 homolog; MAL: Myelin and lymphocyte-associated protein; MCTs: monocarboxylate transporters; MDS: molecular dynamic simulation; mGluR5: metabotropic glutamate receptor 5; MOR: mu opioid receptor; MST: microscale thermophoresis; Na,K ATPase: sodium, potassium ATPase pump; ND: not determined; NDFIP 1/2: NEDD4 family interacting protein 1/2; NEDD4: E3 ubiquitin protein ligase NEDD4; NHERF1: Na+/H+ Exchanger Regulatory Factor; NKCC1: Na-K-Cl cotransporter 1; N-term: N-terminus; PDA: Pull-down assay; PPI: protein-protein interaction; PP1c: Protein Phosphatase 1 Catalytic Subunit; RVD: regulatory volume decrease; SILAC-IP-MS: Stable Isotope Labeling by Amino acids in Cell culture (SILAC)-immunoprecipitation coupled to mass spectrometry analysis; Sipa1l1: signal-induced proliferation-associated 1 like 1; SK3: small-conductance potassium (SK3); SNX27: sorting nexin 27; Spa1: signal-induced proliferation-associated gene-1; SUR1: Sulfonylurea receptor 1; TM2: transmembrane domain 2; TRPM4: Transient Receptor Potential Cation Channel Subfamily M Member 4; TRPV4: Transient Receptor Potential Cation Channel Subfamily V Member 4; µAP 2/3: clathrin assembly protein complexes 2 and 3 µ subunit; USP4: ubiquitin-Specific Peptidase 4; UTA1: urea transporter A1; OVL: overlay assay; YTH: yeast-two-hybrid screening.

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
