# Peer review of "Insight into the Mammalian Aquaporin Interactome"

_ijms, 2022, doi:10.3390/ijms23179615_

Round 1

Reviewer 1 Report

In the manuscript “Insight into the mammalian aquaporin interactome”, the authors proposed to discuss the current knowledge on aquaporins (AQPs) interactomes and address the molecular basis and functional significance of these protein-protein interactions in health and diseases. The paper is of great interest and presents a compilation of information that will be very valuable for those working in AQPs. Nevertheless, I think that there are some relevant improvements that can be done. The authors should also speculate a bit regarding novel functions for AQPs and their possible functional partners. New research lines could be discussed.  Overall the paper is excellent but needs some minor work before it can be considered for publication.  

Specific comments:

1. When referring AQPs localization and physiological roles, some hot topics of research are neglected including reproductive tissues. It can be interesting to include this information throughout the paper.

2. The manuscript is excellent but some future perspectives would be relevant to discuss. In addition, the authors should also comment on new interactomes proposed by others and the possible path for the years to come.

Author Response

We thank the reviewer for the constructive comments.

  1. When referring AQPs localization and physiological roles, some hot topics of research are neglected including reproductive tissues. It can be interesting to include this information throughout the paper.

Concerning AQPs localization and physiological role, we only provided a brief summary for each AQP as the focus of the review concerns AQP protein-protein interactions. Therefore, we mainly discussed the physiological aspects in relation to the AQPs interactome. We further developed aspects related to the reproductive tissue (lines 737-7309: ‘AQP9 is the most abundant AQP expressed in male reproductive tract. During spermatogenesis, the volume modification of differentiating germ cells is predominantly due to the osmotically-driven fluid efflux through AQPs.’

  1. The manuscript is excellent but some future perspectives would be relevant to discuss. In addition, the authors should also comment on new interactomes proposed by others and the possible path for the years to come.

As requested, we discuss the future perspectives, new interactomes and possible paths for the years to come (lines 793-806):

“In the future, sequence-based prediction of protein-protein interaction, bioinformatic compilation of protein-protein interactome, spatial interactome, live-imaging of protein-protein interactions, biophysical and biochemical methods, new photoproximity protein profiling interaction methods, mass spectrometry-based proteomic, thermal proximity coaggregation profiling are anticipated to further uncover the spatial and temporal AQPs interactome that will contribute to more in-depth understanding of physiological and pathophysiological molecular mechanisms. Moreover, structures of AQP complexes of medical relevance are highly interesting for drug design purposes. Here, the development and resolution revolution of single-particle cryo-electron microscopy for membrane protein structural studies is likely to play an important role [185]. Given that structure-based drug design targeting individual AQPs remains challenging despite intense efforts [186–190], switching the focus to the AQPs interactome may offer new therapeutics avenues for treatment of human disease states in which AQP play an important role.”

Reviewer 2 Report

The manuscript is a comprehensive review describing the reported AQPs-protein interactions (interactome), addressing their molecular basis and functional significance for health and disease. The manuscript is very well written and contains a vast list of references. The schematic figures facilitate reading and add value to the manuscript.

A few minor comments may improve the manuscript:

1 – Abstract and Line 23: “The main function of AQPs consist into facilitating osmotically-driven water flux across 13 biological membranes”. Knowing that several AQPs also transport small solutes (glycerol, urea), important for cell physiology, can water transport be considered “the main function”?

2 – line 65-66: AQP11 deserves attention (doi 10.1016/j.redox.2022.102410).

3 – line 76 (legend Fig 1B): “channel through the middle of the tetramer is indicated with an X.” The X is not visible.

4 – Line 377: typo “tssue”

5 – Figure 5: AQP3 and AQP9 expression/function in adipocytes are controversial. A recent study found no evidence of functional overlap between AQP3/AQP9 and AQP7 in human or mouse white adipose tissue (doi 10.1007/s00418-022-02090-4).

Author Response

We thank the reviewer for the constructive comments.

 1 – Abstract and Line 23: “The main function of AQPs consist into facilitating osmotically-driven water flux across 13 biological membranes”. Knowing that several AQPs also transport small solutes (glycerol, urea), important for cell physiology, can water transport be considered “the main function”?

We thank the reviewer for their positive and constructive comments.

We modified the sentence as follows in Abstract - lines 13-14 ‘AQPs facilitate osmotically-driven water flux across biological membranes and, in some cases, movement of small molecules (such as glycerol, urea, CO2, NH3, H2O2).’.

We modified the sentence as follows in Introduction -lines 23-26, ‘Aquaporins (AQPs) are a family of transmembrane water channels expressed in all living organisms facilitating osmotically-driven water flux across biological membranes and, in some cases, movement of small solutes (such as glycerol, urea, CO2, NH3, H2O2) [1,2].’

2 – line 65-66: AQP11 deserves attention (doi 10.1016/j.redox.2022.102410).

Lines 65-66: “To the best of our knowledge, no interaction partners have been identified for AQP10-AQP12, therefore these are not included in the discussion below.”

In doi 10.1016/j.redox.2022.102410, no interacting partner of AQP11 has been identified, so that we do not understand why the reviewer pointed out this reference to us in relation to lines 65-66. However, we modified a paragraph (lines 26-32) in the introduction as follow to add a couple of lines regarding the role of AQP11 as a peroxiporin (lines 30-32).

‘In mammals, 13 different AQPs have been identified that can be subdivided into three groups: classical AQPs permeable to water (AQP0, AQP1, AQP2, AQP4, AQP5, AQP6, AQP8); aquaglyceroporins permeable to water as well as to glycerol and small solutes (AQP3, AQP7, AQP9, AQP10); and unorthodox AQPs (AQP11 and AQP12) with lower sequence homology [1,3]. While solute selectivity of the two latter remains to be conclusively established, AQP11 has been suggested to facilitate transport of glycerol [4] and H2O2  across the endoplasmic reticulum (ER) membrane [5].’

3 – line 76 (legend Fig 1B): “channel through the middle of the tetramer is indicated with an X.” The X is not visible.

We thank the reviewer for pointing out this missing X in Figure 1B. Accordingly, Figure 1B was modified to make the X visible in the middle of the tetramer.

4 – Line 377: typo “tssue”

We corrected typos within the entire manuscript.

5 – Figure 5: AQP3 and AQP9 expression/function in adipocytes are controversial. A recent study found no evidence of functional overlap between AQP3/AQP9 and AQP7 in human or mouse white adipose tissue (doi 10.1007/s00418-022-02090-4).

We thank the reviewer for bringing this recent paper to our attention. We discussed the findings of that paper in lines 439-447 and in Figure 5 -  lines 462-463.

Lines 439-447: “It should be noted however that the expression and function of AQP3 and AQP9 in human adipocytes is controversial. A recent single-cell analysis of human adipose tissue confirmed the expression of AQP7 in human mature adipocytes, but the expression of AQP3 was identified in a low number of preadipocytes and mature adipocytes, AQP9 expression was shown in visceral adipose tissue progenitors, and AQP10 expression could not be detected [109]. In addition, immunohistochemical labeling of AQPs in human adipose tissue suggests no functional overlap between AQP3/AQP9/AQP10 and AQP7 in human or mouse white visceral adipose tissue [109], as opposed to mouse 3T3-L1 adipocytes [101].”

Figure 5 - Lines 462-463: ‘AQP3 and AQP9 may only play a role in mouse 3T3-L1 adipocytes [101], but not in human or mouse white adipose tissue [109].’